# First Evidence of Thalassochory in the *Ficus* Genus: Seed Dispersal Using the Kuroshio Oceanic Current

**DOI:** 10.3390/plants13101398

**Published:** 2024-05-17

**Authors:** Shin-Hung Pan, Ying-Hsuan Sun, Hsy-Yu Tzeng, Lillian Jennifer Rodriguez, Anthony Bain

**Affiliations:** 1Department of Forestry, National Chung-Hsing University, 250 Kuokwang Road, Taichung 40227, Taiwan; 2Institute of Biology, National Science Complex, College of Science, University of the Philippines, Diliman, Quezon City 1101, Philippines; 3Department of Biological Sciences, National Sun Yat-Sen University, 70 Lienhai Rd., Kaohsiung 80424, Taiwan; 4International Ph.D. Program for Science, National Sun Yat-Sen University, 70 Lienhai Rd., Kaohsiung 80424, Taiwan

**Keywords:** ddRAD, *Ficus*, Kuroshio current, Mearns fig, population genetics, single-nucleotide polymorphism, thalassochory

## Abstract

Aim: Plants distributed between southern Taiwan and the north of the Philippines are spread among numerous small islands in an area crossed by the powerful Kuroshio current. Oceanic currents can be effective seed-dispersal agents for coastal plant species. Moreover, the Luzon Strait is an area prone to tropical cyclones. The aim of this study is to look at the dispersal capability of an endangered coastal plant species, the Mearns fig (*Ficus pedunculosa* var. *mearnsii*), using both experimental and population genetics methods. Location: Southern Taiwan, the Philippines, and the islands between Luzon and Taiwan Island. Methods: This study combined two types of analysis, i.e., buoyancy experiments on syconia and double digest restriction-associated DNA sequencing (ddRAD), to analyze the population genetics of the Mearns fig. Results: We first discovered that mature Mearns fig syconia could float in seawater. They have a mean float duration of 10 days to a maximum of 21 days. Germination rates varied significantly between Mearns fig seeds that had undergone different durations of flotation treatment. Population genetic analysis shows a high degree of inbreeding among various Mearns fig populations. Moreover, no isolation by distance was found between the populations and individuals. Main conclusions: From our analysis of the genetic structure of the Mearns fig populations, we can clearly highlight the effect of the Kuroshio oceanic current on the seed dispersal of this fig tree. Comprehensive analysis has shown that Mearns fig seeds are still viable before the mature syconium sinks into the seawater, and so they could use the Kuroshio Current to float to the current population locations in Taiwan.

## 1. Introduction

The publication of theories of island biogeography [1] had permitted us to understand the ecosystems on islands and how they were different from the mainland, but more questions were raised about how the plants could reach these far-away islands. To reach oceanic islands, plants have different methods, such as using animals to bring seeds to new areas (zoochory) or abiotic ways such as wind (anemochory) or seawater (thalassochory). Unintuitively, examples from different archipelagos show that seeds with adaptations for long-distance travel are not necessary to travel long distances [2,3] and differ in their specializations: In the Azores, Portugal, the species with seeds that are adapted for thalassochory were overrepresented [2], whereas in the Galápagos Islands, Ecuador, the same adaptation represented about 19% of the native plants, and 36% of the seeds have no specific adaptations [3]. 

Thalassochory is unknown in the *Ficus* genus, but potamochory (freshwater hydrochory) has been described many times for riparian *Ficus* species [4,5,6]. The fig trees (*Ficus*, Moraceae) are known to be an important source of food for many animals and, thus, a zoochorous genus in its vast majority [7,8,9]. Nevertheless, riparian members of the genus would have floating infructescences that can disperse via water, and adaptations such as spongy tissue in figs have been observed [5,10,11]. Moreover, these riparian species are often still zoochorous but dispersed by less known vertebrates: fishes [12,13]. In any case, the colonization of new territories, such as islands, is difficult for *Ficus* as they rely on their mutualistic wasp for pollination [14]. Indeed, each *Ficus* species appears to have at least one specific pollinating fig wasp species, with which they form a symbiotic and obligate pollination nursery mutualism [15,16,17,18]. As pollen-carrying female fig wasps have a life span of only one to two days [18,19], the vast majority of fig wasps will seek nearby fig syconia for egg-laying/pollination within their life cycle [20]. Female fig wasps can migrate long distances through air currents in search of suitable fig syconia [21,22]. For a *Ficus* species to successfully colonize an area, grown fig trees need to be visited by female pollinating wasps that can successfully establish their progenies in the new habitat. 

The Mearns fig (*Ficus pedunculosa* Miq. var. *mearnsii* (Merr.) Corner) can be found on uplifted coral reef shores in the south and east of the Taiwan Islands and the outlying islands (Orchid Island and Green Island), as well as the Batanes Islands, the Babuyan Archipelago, and the northernmost coast of Luzon Island in the Philippines [23,24], Figure 1. *Ficus pedunculosa* var. *mearnsii* individuals live in a very harsh environment with direct and frequent exposure to seawater and dry, sparse soil [25]. Due to the increase in the frequency and power of extreme meteorological events, the Mearns fig numbers are low—less than 2000 individuals in a few hotspots [26]—and thus have been put on the list of vulnerable species [27]. As the data about the Mearns figs in Taiwan started to increase in quality, a few questions emerged after noticing that their distribution is extremely unintuitive, as the species has not colonized the entire Taiwan coastline, even some areas of uplifted coral reef shores, but it is present on both the Orchid and Green Islands. The first question was as follows: (1) Can the figs of *F. pedunculosa* var. *mearnsii* float? And if they can, (2) how long can they float? Then, if they can float in seawater, another question raised about the effect of seawater on seeds is: (3) Can they germinate? And (4) germinate like seeds that have not been in contact with seawater? At last, we wanted to know if (5) there is any gene flow between these small fig tree populations with tens of kilometers of sea between them and where a strong oceanic current passes through, i.e., the Kuroshio current. To answer these questions, we used seawater to simulate fig flotation and seed germination experiments and carried out double digest restriction-associated DNA sequencing (ddRAD) to analyze the genetic structure of seven Mearns fig populations in Taiwan and its outlying islands.

## 2. Material and Methods

Mearns fig (*Ficus pedunculosa* Miq. var. *mearnsii* (Merr.) Corner) is a dioecious plant from the family Moraceae [10,28] and appears as a prostrate shrub to a multi-branched erected shrub (Figure 1c). Mearns fig is located on uplifted coral reef shores in southern and southeastern Taiwan (Figure 1b), the outlying islands (Orchid Island, Green Island) of Taiwan, and in the Philippines on the Batanes Islands, the Babuyan Archipelago, and the northernmost coastal area of Luzon [23,24]. Mearns figs have been found only on this kind of rock, from a few meters to 30 m away from the sea (Figure 1b,c). The pollinating wasp associated with the plant is *Blastophaga pedunculosae* [29], while the only known non-pollinating wasp belongs to the genus *Apocrypta* [30].

### 2.1. Female Syconium flotation and Seed Germination Experiments

For this study, 49 mature female Mearns fig syconia were collected from Kenting and Jialeshui (Figure 1a) in July 2023. Five mature syconia were randomly selected as the control group. The remaining mature syconia were placed in a 14 cm × 23.5 cm × 23.5 cm glass tank filled with seawater (collected from the Kenting shore) at room temperature (28 °C–30 °C). Then, an “hang-on-back” aquarium filter was installed to simulate the constant state of disturbance of the sea surface. The figs stayed in the water until they sank.

Nevertheless, during the flotation experiment, five syconia were randomly removed on days 3, 6, 9, 12, and 15. Figs with fully developed drupelets (seeds) were dissected, and the drupelets were taken out of the syconium. All the drupelets of each fig were placed in the Petri dishes with a moistened paper towel and then placed in a room with sunlight for germination in the control groups. Seeds were determined to have germinated when a radicle began to show outside the drupelet. The number of days taken for the first seed to germinate in the various syconium experiments as well as the total number of germinated drupelets in each group were noted. The mean germination time, germination rate index, and final germination percentage [31,32] were then calculated; syconia that had not developed fully and those empty of seeds were excluded. The germination rate index (%/day) was calculated according to the following formula: GRI = $\sum_{t=1}^{n}\left(G_{t}/t\right)$ where G_t_ is the percentage of germination occurring during time interval t, and n is the duration of incubation (days). Higher GRI values indicate higher and faster germination [31].

The germination rates of the syconia at various flotation time points were arcsine-square-root transformed (y* = arcsin$\sqrt{y}$) prior to analysis, and the R software [33] was used to carry out analysis of variance (ANOVA) and Fisher’s protected least significant difference test (LSD test) to compare the correlation coefficients of seed germination between different flotation time points.

### 2.2. Sampling and DNA Extraction

In this study, Mearns fig populations from seven areas, namely Jialeshui, Kenting, Eluanbi (Southern Taiwan), Shiauyeliou, Sansiantai (Eastern Taiwan), Orchid Island, and Green Island, were collected (Figure 1, Table 1 and Table 2). Fifteen individuals of each sex were collected from each population, totaling seven populations and 210 individuals. The genomic DNA of the samples was extracted from mature leaves using the GeneMark Plant Genomic DNA Purification kit (GeneMark Technology Co., Ltd., Tainan, Taiwan). The quality and quantity of the extracted DNA were determined by a Nanodrop UV spectrophotometer (Thermo, Waltham, MA, USA) and stored in the −20 °C freezer.

### 2.3. ddRAD Library Preparation and Sequencing

The adapters and primers were designed based on Shirasawa et al. [34,35]. Genomic DNA samples of *Ficus pedunculosa* var. *mearnsii* were digested with restriction enzymes PstI and MspI (FastDigest Restriction Enzymes, Thermo Scientific™). Moreover, 0.8 μg of DNA, 2.5 μL of PstI, 2.5 μL of MspI, and 5 μL of 10× FastDigest Buffer were mixed, then deionized water was added to reach a final volume of 50 μL. The reaction mix was incubated at 37 °C overnight to allow for enzyme digestion. Electrophoresis was used to confirm whether the DNA had been digested into small fragments. Finally, AMPure XP beads (Beckman Coulter, Brea, CA, USA) were used for clean-up by removing the restriction enzyme. 

The adaptor ligation was conducted using 25 μL of the restriction enzyme digestion product, 2 μL of PstI adapter (10 μM), 2 μL of MspI adapter (10 μM), 5 μL of 10× T4 ligation buffer, and 2 μL of NxGen^®^ T4 DNA ligase (Lucigen Corporation, Middleton, WI, USA) and incubated at 23 °C for 1 h and 15 °C for 2 h using a high precision temperature gradient reactor (TProfessional 96 Thermocycler, Biometra, Gottingen, Germany). Fragments size of 300–800 bp of the ligated products were selected using AMPure XP beads (Beckman Coulter, Brea, CA, USA) and followed by PCR amplification using primers and the JMR hot start PCR mix kit (JMR Holdings, West Midlands, UK) with the following conditions: 94 °C for 9 min, followed by 35 cycles of 94 °C for 30 s, 62 °C for 30 s, and 72 °C for 1 min, and then 72 °C for 10 min. The library quantities and qualities were evaluated using the Qubit dsDNA HS Assay Kit and qualitative analysis using the BiOptic Qsep400 and High Sensitivity (N1) Cartridge Kit (Fort Wayne, IN, USA). Libraries were sequenced using the Illumina Hiseq 4000 (San Diego, CA, USA) to perform 150 bp paired-end sequencing. The sequencing service was provided by Genomics BioSci and Tech Co., Ltd., Taiwan, China.

### 2.4. ddRAD Data Processing

FastQC v0.11.9 [36] was used to evaluate the quality of the sequencing results. STACKS (https://catchenlab.life.illinois.edu/stacks/ accessed on 9 May 2024) software was used for de novo assembly of all fragments into the reference sequence, and preliminary screening of the raw sequence data was performed using ‘process_radtags–c–q’. Using ustacks, the minimum number of reads to form a stack was set to be 3, and the minimum mismatch for locus formation was set to 2 for fragment assembly and clustering. Cstacks was then used to determine and catalog loci, with the minimum mismatch between loci set to 1. Individual SNP genotypes were determined using sstacks; tsv2bam was used to convert the individual SNP genotype into an SNP locus sequence; and gstacks was used for integration to generate the catalog.fasta [37].

The catalog assembled loci sequence from STACKS was used as a reference sequence for ipyRAD (https://ipyrad.readthedocs.io/en/master/ accessed on 9 May 2024) [38] mapping and SNP loci identification for all samples. Only sequence reads that had no more than 5 bases of quality less than Q20 were processed. SNP loci were identified with the following parameters: a minimum permitted deviation of 33, a minimum depth of 6 bases in every locus, and a maximum cluster depth of 10,000 for each sample. A maximum of 25% heterozygous loci were permitted per locus; a maximum of 20 SNPs and 5 indels (insertion/deletion) were permitted per locus.

TASSEL 5 (https://tassel.bitbucket.io/) was used to conduct principal components analysis (PCA) and Tajima’s D calculation using each SNP locus meeting the following requirements: 60+ samples, minor allele frequency (MAF) ≥ 0.1, heterozygosity ≤ 0.05, and removing minor SNP status as a threshold for screening [39,40]. The PCA results were plotted using R/ggplot2 (ver. 3.3.6). R/poppr was used to perform minimum spanning network (MSN) analysis on all samples [41]. Twisst, the open script setting in Python 3.8 (https://github.com/simonhmartin/twisst accessed on 9 May 2024), was used to set the window size to 500 bp and each window set to require at least 25 SNPs for the calculation of nucleotide diversity (π), genetic distance (D_XY_), and F_ST_ [42,43,44,45]. R/genepop was used to calculate intra-population genetic difference (F_IS_), and private alleles were used to calculate gene flow (Nm) [45,46,47]. SNPs that were selected after TASSEL 5 screening were used to conduct sparse non-negative matrix factorization (sNMF) using the sNMF command of R/LEA to analyze the genetic structural composition with the following parameters: K = 2 to 10, correction parameter (alpha) of 100, 1,000,000 iterations, and 20 repeats for each group [48,49,50].

The program LDNe [51] was used to calculate the effective population size (N_e_) and linkage disequilibrium evaluation. The prerequisites for LDNe estimation of effective population size are a neutral locus, close relationships between populations, non-overlapping generations, and that they are not easily affected by high migration rates [52]. These hypotheses are similar to the situation in Mearns fig.

## 3. Results

### 3.1. Female Syconium Flotation and Seed Germination Experiments

The syconia could float in seawater for 7 to 21 days, with a mean flotation duration of 9.9 ± 7.0 days. On day 9, more than 50% of the syconia were still floating (Figure 2a). On day 15, only one syconium continued to float, and it sank on day 21. The external integument of the syconia started to decompose around days 3–5. At this time, the syconium became more buoyant and would only start to sink once the walls had completely decomposed.

For the germination experiments, the number of recorded seeds per fig ranged from 22 to 119, and the mean number of seeds was 62.5 ± 24.2. The seeds from the control group, which was not immersed in seawater, started to germinate after 5 days, with a mean germination duration of 7.0 ± 1.2 days, a seed germination rate of 14.5 ± 2.63%/day, and a final germination rate of 96.2 ± 2.99%. In the flotation treatment groups, the mean germination duration was 12–17.9 days, the seed germination rate was 5.5–9.4%/day, and the final germination rate per fig was 70.23–96.04% (Figure 2b–d, Table 3).

ANOVA results showed that the mean duration of germination and seed germination rates showed significant differences among Mearns fig seeds that had undergone differing durations of flotation treatment, but there were not significant differences between the groups in terms of final germination rate (Table 3). LSD test results showed that there were differences in mean germination duration and seed germination rate between the different treatment groups and the control group, while the differences in the final germination rate were not significant between groups (Table 4).

### 3.2. Mearns Fig Genetic Structure

The total DNA sequencing data size in this study was 161.8 Gbp, with a total of 1,071,811,586 reads. After STACKS was used to cluster the reads from 210 samples, a total of 1,229,510 loci were assembled and integrated into catalog.fa. The mean coverage of each locus was 15.6x (4.2x–98.5x). Using catalog.fa as a reference sequence, a total of 191,181,944 reads were retained for mapping after ipyRAD screening. The mapping results retained 84,930 loci, with a mean length of 225 bp, a coverage depth of 6.33x, a simplified genome size of 19,153,885 bp, and a total of 227,645 SNPs identified (Table 5).

After TASSEL 5 filtering, 232 SNPs were retained for PCA of 210 Mearns figs. PC1-PC5 explained 56.7% of genetic variation, while PC1, PC2, and PC3 explained 29%, 12.8%, and 8.1% of genetic variation, respectively. Pairwise scatter plots of PC1, PC2, and PC3 were produced (Figure 3). The 95% confidence ellipse of the seven populations contained overlapping samples common to all populations, but there were significant differences in the distribution sizes and slopes of these ellipses. In these different scatter plots (Figure 3), the southernmost locations of Orchid Island and Eluanbi have the smallest surfaces, while Sansiantai, Shiauyeliou, and Green Island, which are located in the north, and Kenting, located in the west, have the largest surfaces. 

sNMF analysis found that the cross-entropy at K = 5 was the lowest, showing that K = 5 is the optimal number of genotype source clusters (Figure 4). The results in Figure 5 show that most of the individuals (represented as columns in Figure 5) consist of one genotype source. All genotype sources can be found in Sansiantai individuals, and the populations from Shiauyeliou and Sansiantai, which are the two East Taiwan populations, are the only ones having the green genotype (Figure 5). The proportion of the four other genotype sources is extremely similar between the two outlying islands, the Orchid and Green Islands (Figure 5 and Figure 6). On the other hand, the Jialeshuei and Eluanbi populations have a higher proportion of the yellow genotype source than the outlying island populations. They are also quite similar in genotype proportion (Figure 6). Lastly, the Kenting population is different from all other populations, with an important proportion of the light blue genotype source reaching about two-thirds of the total (Figure 5 and Figure 6).

The F_IS_ of Mearns fig populations ranged from 0.181 to 0.432 (Table 5); the lowest and highest F_IS_ values were both in East Taiwan (Sansiantai and Shiauyeliou, respectively). The values of π ranged from 0.027 to 0.096, with the lowest being Shiauyeliou and the highest being Kenting. Tajima’s D range was −4.476–−1.303, with all values being negative, i.e., the lowest was from Green Island and the highest was from Sansiantai. The π values (0.0274–0.0290) of the three populations located in the northernmost three groups of Sansiantai, Shiauyeliou, and Green Island were significantly lower than those of other populations (Table 5). The Pearson coefficient (*r*) of π and F_IS_ in various populations was only −0.0913 (*p*-value = 0.8298), indicating that the two sets of data were weakly correlated.

The D_XY_ range was 0.0223–0.166, with the genetic distance between the Sansiantai and Shiauyeliou populations being the closest. On the other hand, the Green Island and Jialeshui populations had the highest values (Table 6). F_ST_ values had a greater range than D_XY_ values, ranging from 0 to 0.836. The 0.836 F_ST_ value is attributed to the distance between Green Island and Sansiantai, whereas three pairs of populations have a F_ST_ value of zero (Table 6). The F_ST_ values between Green Island and other populations have high genetic differentiation (average: 0.582). Contrarily, the highest F_ST_ value between the three southern populations (Kenting, Jialeshui, and Eluanbi) was only 0.0323 (Table 6), while the F_ST_ value between the two eastern Taiwan populations (Sansiantai and Shiauyeliou) was zero. The F_ST_ of Orchid Island populations and eastern and southern Taiwanese populations showed little to no genetic difference (0–0.0689). Nm ranged from 0.167–0.471, with all values less than 1. The lowest Nm was between Kenting and Jialeshui, and the highest was between Green Island and Eluanbi (Table 6).

The results of LDNe estimation of the effective Mearns fig population size (N_e_) (Table 7) showed that when all samples were viewed as a single population, the N_e_ was 1.5 (95% CI: 1.4–1.5). All but one of the N_e_ values are at 5 or below, but the Eluanbi N_e_ value is outstanding at 104.6 (Table 7).

## 4. Discussion

Our study results show that the female figs of the Mearns figs can float for up to 21 days, with an average of about 10 days. Moreover, the genetic structure of the studied fig tree follows their geographical locations and grouping: The southern three populations are genetically close, as are the two eastern populations, with the two island populations being slightly apart from each other and the Taiwan Island populations. The different analysis methods show that the eastern populations are strongly grouped together in the Bayesian clustering figure (Figure 5), and the population pair F_ST_ values are notably higher for the Green Island population.

### 4.1. Buoyancy and Seed Dispersal

The seed dispersal of *Ficus* species is known to be a good example of zoochory, as more than 1200 species of vertebrates have been documented feeding on figs [8]. Moreover, the *Ficus* genus species have a very wide range of habitats and also live in riparian forests [4,5,10]. Few species of freshwater fish have been observed eating figs in Costa Rica [6,13,54], but the actual number may be much higher [55]. On the other hand, the relationship between seawater and fig dispersal is totally unknown, with, for example, only a few *Ficus* species being very coastal in Malesia over more than 350 species: *Ficus pedunculosa*, *F. deltoidea*, *F. edanoi*, *F. opposita*, or *F. concinna* [10]. As far as our knowledge goes, we did not find any publication describing the seed dispersal of fig seeds using sea currents, even for coastal *Ficus* species.

Seed dispersal by oceanic currents (also called thalassochory) is rare in angiosperms and is considered one of the main mechanisms to colonize oceanic islands [2], also called long-distance dispersal. In order to use oceanic currents for seed dispersal, plants need to fulfill two main requirements, i.e., (1) they need to live in proximity to the seashore for their seed-bearing fruits to reach seawater; and (2) their seeds and/or fruits have to be buoyant in order to reach other landmasses. Thalassochory is an overlooked way for the plants to disperse their spores over long distances and may be more common than currently known for over extreme dispersal distances [56] and more locally [57]. The studied species, *F. pedunculosa* var. *mearnsii*, fulfills the two above requirements for thalassochory. Indeed, we show that the seeds are buoyant, and the distribution of these trees is in thin areas extremely close to the Taiwan seashore. Moreover, the genetic population structure is also hinting towards a seed dispersal method using the sea current.

Nevertheless, another matter is important to discuss, i.e., the seed dispersal by animals (zoochory). Indeed, *Ficus* trees are known to be mainly dispersed by animals [8], with a large part of them being birds. Unfortunately, *Ficus pedunculosa* is not mentioned in Shanahan et al.’s work [8], but the closest relative, *F. deltoidea,* is dispersed by a monkey. Moreover, none of the authors have observed animals eating the figs of the Mearns fig, even if rare-bitten figs have been observed. Figs of this size eaten by birds tend to be swallowed in one bite [58]. Therefore, the Mearns fig zoochory appears unlikely. Secondly, zoochory would have smoothed the genetic structure within the populations. For example, the south or east populations would have been more homogeneous, with a gene flow mostly carried out under bird zoochory. Each of these populations is geographically close to the others, and birds feeding specifically on these figs would have flown easily between these locations for food. For these reasons, the zoochory hypothesis is not the main hypothesis for the Mearns fig gene flow. 

### 4.2. Population Genetic Structure and the Kuroshio Current

The trajectory of the Kuroshio current has been described with extreme details [59]. It passes through the overall distribution range of the Mearns fig (Figure 5).

Therefore, the influence of the Kuroshio current can be seen through the results of our genetic analyses and the distribution of the Mearns fig populations. There are many hints that lead us to link the genetic structure of the Mearns figs to the Kuroshio current influence.

First, the Green Island population is different from the other populations regarding the F_ST_, D_XY_, and Nm values, which are higher on average than other population pairs. Interestingly, the Green Island population is not the most isolated of the studied populations (Sansiantai is an average of 108 km away from other populations), but it has genetic indicators that set it apart from the other populations. This situation can be explained by the position of the island in the Kuroshio oceanic current. Indeed, the speed of the current around the island is the fastest around Taiwan Island, and it is moving northeast, parallel to the Taiwan East Coast [59]. Considering that fact and the geographical location of the island, it is easy to comprehend that the Mearns fig population there can only receive newcomers (still, the Nm values are low), but the seeds traveling from the island will only go northwards and away from the other populations. Moreover, the average current speed in the south of Taiwan is 50 cm·s^−1^ [59], which is about 1.8 km·h^−1^. At this average speed, a seed will travel more than 400 km in 10 days.

Second, geographically, the Green Island population is linked with the two eastern populations, Shiauyeliou and Sansiantai, making the three northernmost populations, and as the Kuroshio current is oriented in the north–south direction, this is an important characteristic. These three populations share the lowest nucleotide diversity π values, which are weakly correlated with the F_IS_ values, leading us to think that the low nucleotide diversity may be due to a low founder effect. Moreover, the two eastern populations form a unique cluster group in the powerful structure clustering analysis that is totally absent from other populations, showing the uniqueness of their genetic diversity among the studied populations. Similarly, with the population of Green Island, the geographic isolation of these two Eastern populations may have led to genetic differences over time.

Third, the Orchid Island and eastern Taiwan populations are closer genetically than they are geographically (F_ST_ and D_XY_ values). This may be due entirely to the pattern of the Kuroshio current: Indeed, in the maps of the Kuroshio current, there is a clear deviation from the strong northward current that is pointing directly at the location of the Shiauyeliou population (see Figure 6; Figures 1 and 11 in [59]). Similarly, the southern populations could be sending seeds via the Kuroshio current to Orchid Island, as the eastern populations could also do. Moreover, the distribution of the Mearns fig populations is not continuous along the Taiwan East Coast from Jialeshui to Sansiantai, but only a few populations (we have sampled all the known populations). It is a hot-spot cluster distribution [28]. The position and existence of the Sansiantai and Shiauyeliou may be due only to the deviation from the Kuroshio current that meets the coastline at the area where the Shiauyeliou population is.

Lastly, the Kenting population is different from the other populations because of its location. It receives the southeast branch of the Kuroshio current that flows from the western coast of Taiwan. Therefore, it is not part of the main current that flows west of the Batanes Archipelago. It may receive no seed from the Batanes but perhaps more from the Babuyan Islands, as the Kuroshio branch that flows southwest of Taiwan Island passes through the Babuyan Islands. According to the structure clustering, the Kenting population is different from the other populations in terms of the proportion of the different genotypes but not genetically differentiated from them (according to the other genetic indices). The Kenting population also has the highest nucleotide diversity in this study.

### 4.3. Overall Genetic Structure

The gene flow of *Ficus* species is known to have two main characteristics: zoochory [8], long-distance pollination events that create genetic distance between populations that is extremely low, and high intra-population genetic diversity [21,22,60,61]. This phenomenon reflects the broad natural geographical spatial distribution and low plant density characteristics of monoecious figs [53,62,63,64,65]. On the other hand, genetic studies of dioecious figs have also shown low genetic differentiation among different populations and high intra-population genetic diversity, with no differences in genetic structure between populations of different sexes [66,67,68]. It has been shown that the reproductive system and the size of the trees greatly affect the dispersing distances of the pollinating fig wasps [69,70,71]. In other words, dioecious *Ficus* species pollen dispersal is shorter than monoecious pollen dispersal. The examples are similar in Taiwan: Three dioecious fig species (*F. erecta* var. *beecheyana* (subg. *Ficus*), *F. benguetensis*, and *F. septica* (subg. *Sycomorus*)) that are widely distributed in Taiwan have very low genetic differentiation [72]. Of these, the southern *F. septica* population showed differences in genetic structure compared with other populations due to the presence of a different species of pollinating wasp in southern Taiwan. The primary geospatial characteristics of these fig plants are low intra-population densities and broad distribution ranges, thereby further demonstrating the effects of wasp dispersal distances on the genetic structure of dioecious plants [20,72,73]. 

However, the results of this study differ from the aforementioned studies. Our results show high inter-population genetic diversity in Mearns fig populations in Taiwan and its outlying islands and low intra-population genetic diversity. F_IS_ results showed that the various Mearns fig populations have a tendency for inbreeding, and the inter-population Nm value was lower than 1, with the highest being only 0.471. This phenomenon suggests that there is isolation in gene flow between all populations or that populations are insufficiently resistant to inter-population genetic differentiation caused by intra-population mutations [74]. This phenomenon is consistent with Tajima’s D of all measured Mearns fig populations being lower than 0, showing that all populations are affected by the founder effect or bottleneck effect [40]. Although the composition of the genetic structure was similar in all Mearns fig populations, the genotypic sources and composition were not the same, and the composition shows a significant uneven distribution in all populations. The tendency toward homogenous genetic structural composition, the low number of individuals per population, and difficulty in expansion all show that genetic drift significantly affects inter-population differentiation in Mearns fig [75]. For founder populations that colonize islands, inbreeding also accelerates genetic drift [76,77,78,79,80]. This is especially true if the number of founders is low or when the population size shrinks, as high inbreeding will increase the number of additive genetic variables, resulting in differentiation to overcome genetic limitations [81,82,83,84,85,86].

The spatial genetic structure of Mearns fig is affected by its hot spot cluster distribution [28]. where the individuals are clustered in some areas with great distances between clusters. As *F. pedunculosa* var. *mearnsii* is a small dioecious species, its pollinating wasps may travel shorter distances than larger-sized fig trees [69,70,71]. Nevertheless, the recovery from the damages caused by typhoons (the names of tropical cyclones in this area) showed that the pollinating wasps are capable of recolonizing an area coming from undamaged areas [26]. The genetic structural characteristics of the Mearns fig show that there has been extremely low gene flow between populations since the founders of the various populations became established on uplifted coral reefs. Mearns fig shows a discontinuous hot spot distribution in Taiwan with no significant differences in gene flow between the various populations in Taiwan and the two outlying islands of Orchid Island and Green Island. Moreover, besides the two Eastern populations, both pollinating wasps and Mearns figs are affected by their geographical locations in an area that is prone to typhoons, which are probably reducing the available genetic diversity by reducing the number of individuals [26], which is particularly striking when the comparison between the actual number of living individuals and the estimated effective population size, N_e_ (Table 7). Although this study lacks data from Luzon, the Babuyan Archipelago, and Batanes, we were unable to confirm the possible original origin of the five genotypes that have migrated to Taiwan. Previous studies have also shown that the genetic structure of sea-floating plant populations is affected by ocean current direction for Northern Kuroshio and Southern Kuroshio [87,88,89,90].

## 5. Conclusions

With these additional Filipino populations, it would have probably made a more complete view of the genetic population of the Mearns Fig. However, the syconium sea flotation simulation, the seed germination experiments, and the population genetic structure results showed that mature Mearns fig syconia are mainly dispersed by the Kuroshio Current in a south-to-north direction, and populations are established in this manner. In the future, Mearns fig populations in Batanes, the Babuyan Archipelago, and northern Luzon of the Philippines could be included in analysis to understand the genetic, geographic, and spatial structural mechanisms by which Mearns fig populations disperse to and colonize Taiwan and its outlying islands and to provide a reference for conserving the genetic diversity of the Mearns fig.

## Figures and Tables

**Figure 1 plants-13-01398-f001:**
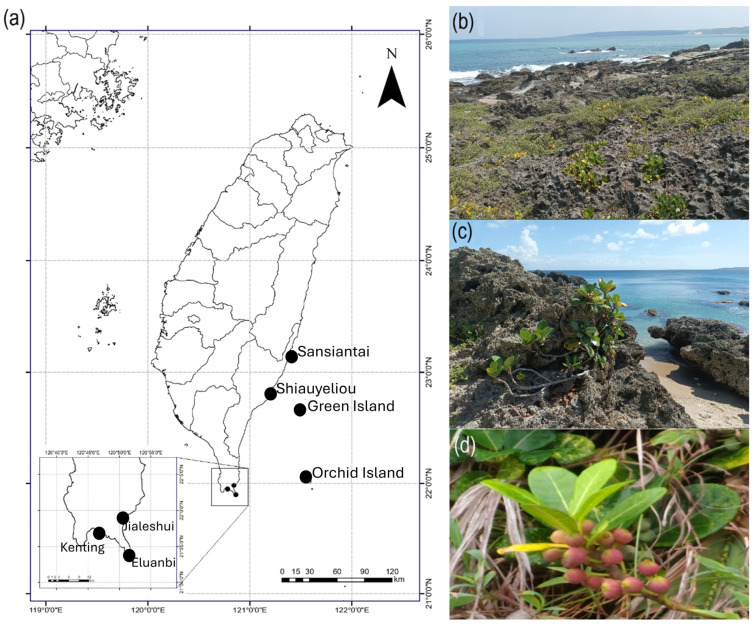
Geographic distribution, morphology, and habitats of *Ficus pedunculosa* var. *mearnsii*. (**a**) Sampling locations in Taiwan. (**b**) Uplifted coral reef at Kenting. (**c**) Typical *F. pedunculosa* var. *mearnsii* individual. (**d**) Figs of *F. pedunculosa* var. *mearnsii*.

**Figure 2 plants-13-01398-f002:**
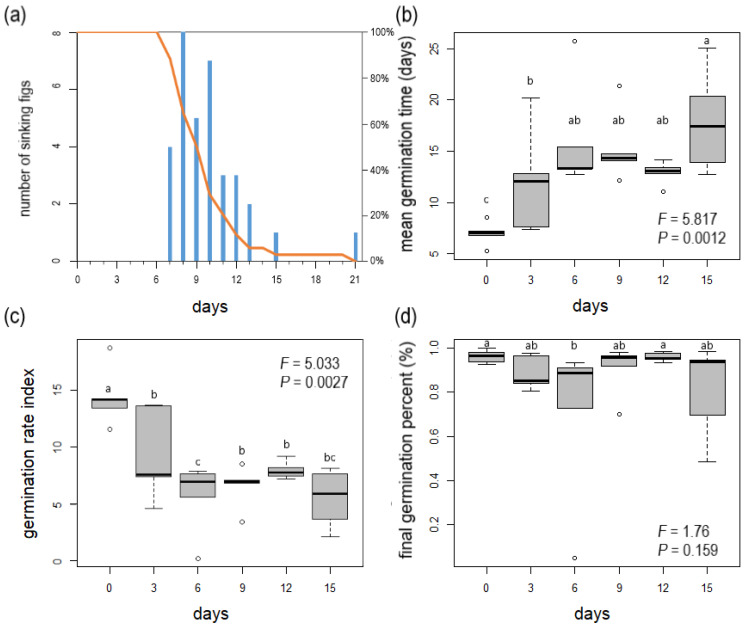
Germination and floating experiments. (**a**) *Ficus pedunculosa* var. *mearnsii* sinking figs (bars) and proportion of remaining floating figs (line). (**b**–**d**) Germination indexes with results of ANOVA analysis for single trait variation of figs.

**Figure 3 plants-13-01398-f003:**
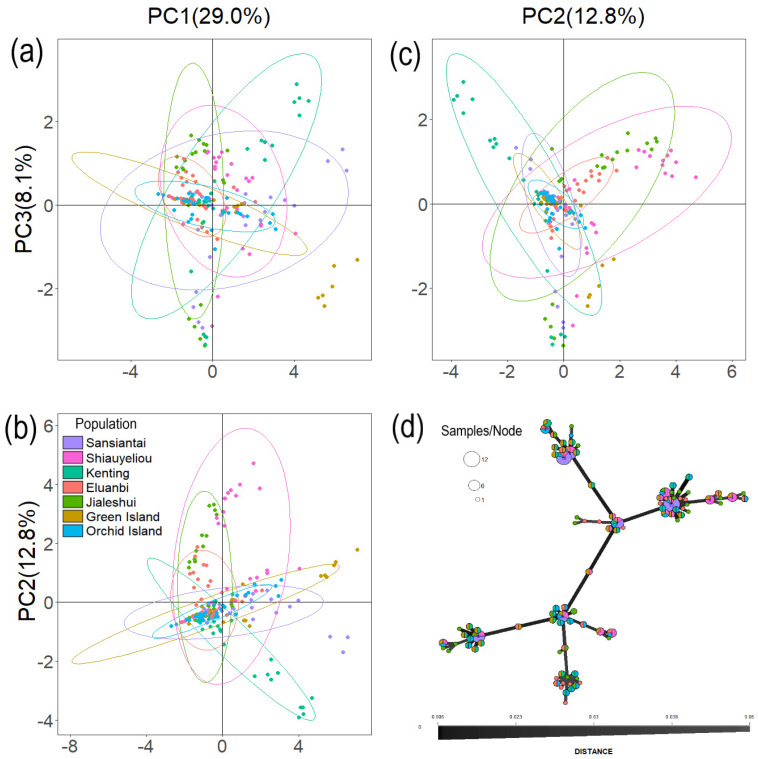
Principal component analysis (PCA) plot with 95% confidence ellipses. PCA results, which include graphs of (**a**) PC3 vs. PC1, (**b**) PC2 vs. PC1, and (**c**) PC3 vs. PC2. (**d**) Minimum spanning network (MSN) analysis showing the genetic distance among individuals within each of the seven populations.

**Figure 4 plants-13-01398-f004:**
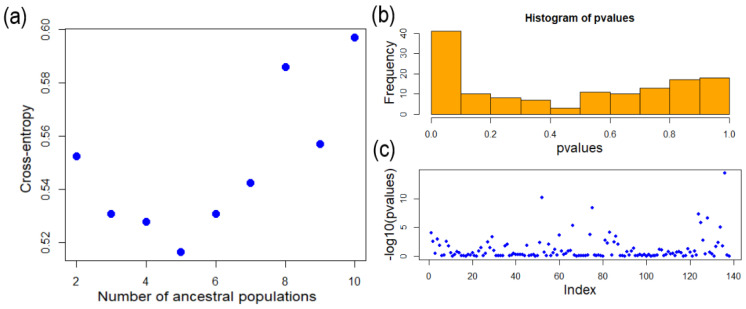
Data from the structure analysis from the R package LEA. (**a**) Cross-entropy of possible K value estimation (K = 2–10) and (**b**,**c**) *p*-values of SNPs when the ancestor number is assumed to be 5.

**Figure 5 plants-13-01398-f005:**
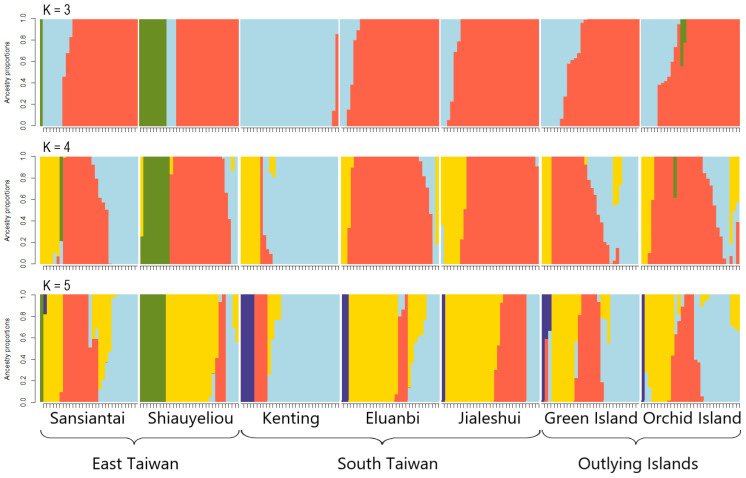
Population structure of the seven *Ficus pedunculosa* var. *mearnsii* populations for K = 3, K = 4, and K = 5. Each individual was represented by a vertical line, with its color indicating its proportion to ancestral populations.

**Figure 6 plants-13-01398-f006:**
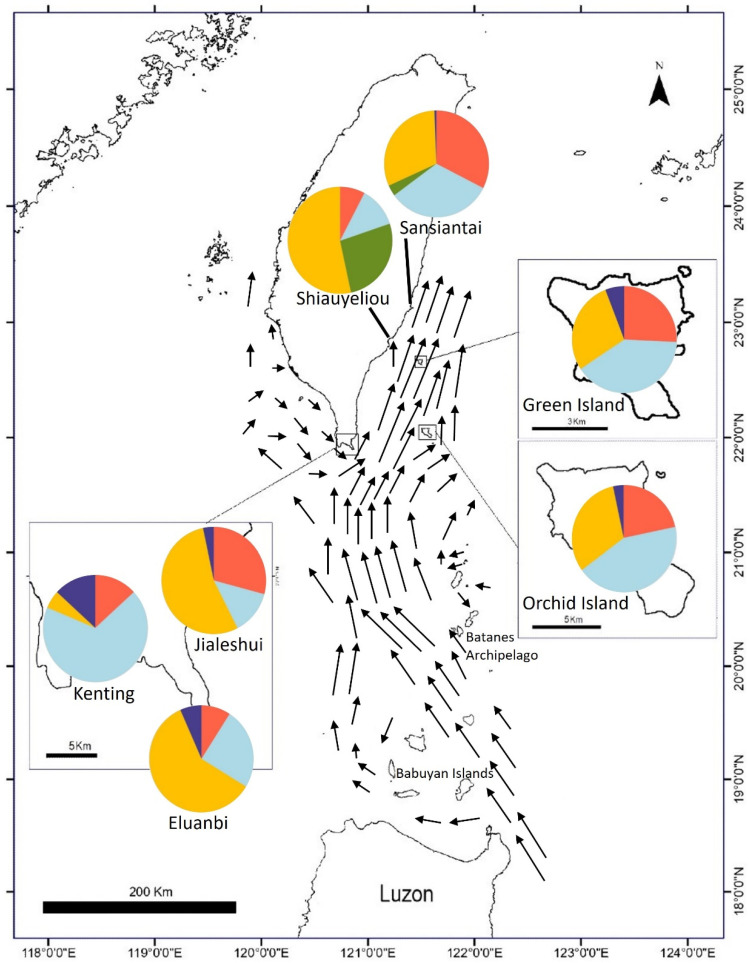
Map of the different genotypes (shown in Figure 5) and Kuroshio current. The arrows represent the current vectors (modified from Liang et al. 2008 [53]).

**Table 1 plants-13-01398-t001:** GPS coordinates of the seven sampling populations.

Distribution	N	GPS Coordinates
*Ficus pedunculosa* var. *mearnsii*	210	
East Taiwan:		
Sansiantai	30	23°12′34″ N; 121°41′23″ E
Shiauyeliou	30	22°79′49″ N; 121°19′76″ E
South Taiwan:		
Kenting	30	21°93′94 N; 120″79°79″′ E
Eluanbi	30	21°89′93″ N; 120°85′7″ E
Jialeshui	30	21°99′41″ N; 120°86′30″ E
Outlying Islands:		
Green Island	30	22°66′18″ N; 121°47′83″ E
Orchid Island	30	22°2′67″ N; 121°56′79″ E

**Table 2 plants-13-01398-t002:** Geographic distance (km) among different *Ficus pedunculosa* var. *mearnsii* populations.

	Sansiantai	Shiauyeliou	Kenting	Eluanbi	Jialeshui	Green Island
Shiauyeliou	42.6					
Kenting	146	104				
Eluanbi	148	106	7.03			
Jialeshui	138	95.5	9.06	10.6		
Green Island	51.8	32.7	107	107	97.5	
Orchid Island	123	93.5	80	75.3	72.7	71.2

**Table 3 plants-13-01398-t003:** The ANOVA results of different seed germination traits.

		Mean Square		
Source	df	MGT	GRI	FGP
treatment	5	0.018 **	0.019 **	0.104
Residuals	24	0.003	0.004	0.059

MGT: mean germination time. GRI: germination rate index. FGP: final germination percent. ** Significant at the 0.01 level.

**Table 4 plants-13-01398-t004:** Comparisons among the number of days of fig floating on the sea surface for different seed germination traits. Data are given as mean ± standard error (*n* = 5). Means within each column followed by the same letter(s) are not significantly different at the 5% level by Fisher’s protected LSD test.

Treatment (Days)	Percentage of Figs Still Floating at the End of the Treatments (%)	MGT * (Days)	GRI *	FGP * (%)
0	100%	6.98 ± 1.18 ^c^	14.47 ± 2.63 ^a^	96.23 ± 2.99 ^a^
3	100%	12.00 ± 5.19 ^b^	9.38 ± 4.10 ^b^	88.83 ± 7.75 ^ab^
6	100%	16.10 ± 5.49 ^ab^	5.64 ± 3.18 ^b^	70.23 ± 37.47 ^b^
9	50%	15.32 ± 3.54 ^ab^	6.56 ± 1.90 ^c^	90.47 ± 11.69 ^ab^
12	12%	12.89 ± 1.15 ^ab^	7.97 ± 0.79 ^b^	96.04 ± 2.10 ^a^
15	3%	17.88 ± 5.01 ^a^	5.50 ± 2.60 ^bc^	81.02 ± 21.34 ^ab^

* MGT: mean germination time; GRI: germination rate index; FGP: final germination percent.

**Table 5 plants-13-01398-t005:** Numbers of SNP, inbreeding coefficient value (F_IS_), Tajima’s D value, and nucleotide diversity (π) of each *Ficus pedunculosa* var. *mearnsii* population.

Population	SNPs	F_IS_	Tajima’s D	π
*Ficus pedunculosa* var. *mearnsii*	227,495	0.443	−7.472	0.077
East Taiwan	190,227	0.420	−2.429	0.071
Sansiantai	113,299	0.181	−1.303	0.028
Shiauyeliou	129,315	0.432	−2.736	0.027
South Taiwan	219,548	0.441	−5.189	0.065
Kenting	155,846	0.251	−3.254	0.096
Eluanbi	109,460	0.258	−3.918	0.082
Jialeshui	136,694	0.425	−2.479	0.060
Outlying Islands				
Green Island	52,733	0.239	−4.476	0.029
Orchid Island	105,355	0.277	−2.139	0.087

**Table 6 plants-13-01398-t006:** Genetic differentiation among *Ficus pedunculosa* var. *mearnsii* populations.

		Sansiantai	Shiauyeliou	Kenting	Eluanbi	Jialeshui	Green Island
Shiauyeliou	DXY	0.022					
	FST	0					
	Nm	0.284					
Kenting	DXY	0.031	0.034				
	FST	0.086	0.035				
	Nm	0.372	0.252				
Eluanbi	DXY	0.060	0.068	0.081			
	FST	0.223	0.111	0			
	Nm	0.253	0.254	0.235			
Jialeshui	DXY	0.065	0.057	0.054	0.071		
	FST	0.161	0.103	0.032	0.031		
	Nm	0.178	0.221	0.167	0.269		
Green Island	DXY	0.052	0.056	0.122	0.109	0.166	
	FST	0.836	0.724	0.578	0.369	0.623	
	Nm	0.376	0.376	0.357	0.471	0.298	
Orchid Island	DXY	0.035	0.034	0.069	0.085	0.051	0.109
	FST	0.040	0	0.033	0.069	0.056	0.362
	Nm	0.445	0.278	0.368	0.219	0.184	0.334

**Table 7 plants-13-01398-t007:** Effective population size (N_e_) estimates by LDNe of each *Ficus pedunculosa* var. *mearnsii* population.

	Estimated Number of Individuals in the Wild *	Independent Comparisons	Overall r^2^	Expected r^2^	N_e_	95% CIs	
						Parametric	Jackknife Loci
All		36,688	0.1496	0.0435	1.5	1.4–1.5	1.4–1.5
East Taiwan		2269	0.5895	0.0896	0.3	0.3–0.3	0.3–0.3
Sansiantai	100	171	0.4339	0.1825	0.6	0.4–0.9	0.4–0.9
Shiauyeliou	100	1229	0.4810	0.1083	0.4	0.4–0.5	0.4–0.4
South Taiwan	2000	65,251	0.0450	0.0300	20.0	19.3–20.7	18.8–21.3
Kenting		4496	0.1947	0.1381	2.7	2.4–3.2	2.3–3.3
Eluanbi		51,404	0.0988	0.0959	104.6	73.1–179.6	71.7–188.1
Jialeshui		89,667	0.1658	0.1238	4.7	4.3–5.1	4.1–5.2
Outlying Islands							
Green Island	100	405	0.2844	0.2110	2.1	1.3–5.7	1.3–6.4
Orchid Island	100	1115	0.2823	0.2416	5.0	2.3–14.6	2.3–13.4

* Data from Chiu et al. (2017) [26].

## Data Availability

The data from this study has been uploaded online (www.ncbi.nlm.nih.gov/sra/PRJNA1079886 accessed on 9 May 2024) and can be cited under as SRA data PRJNA1079886 (Temporary Submission ID: SUB14263187).

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
