# Peer review of "First Evidence of Thalassochory in the *Ficus* Genus: Seed Dispersal Using the Kuroshio Oceanic Current"

_plants, 2024, doi:10.3390/plants13101398_

Round 1
Reviewer 1 Report
Comments and Suggestions for Authors
This is a review of Pan et al “first occurrence of thalassochory…” for Plants. This manuscript uses buoyancy experiments combined with seed germination tests to assess whether syconia (figs) of the species Mearns fig could disperse via ocean currents. It then uses ddRAD to analyze the population genetics of the species. I enjoyed reading this. The question is interesting, the MS is for the most part clearly written, and the data well presented. However, I think the manuscript would benefit from a broader view and a clearer assessment of alternative hypotheses of dispersal (both pollen and seeds) and how the population genetic data ties into those alternatives. I also have some suggestions regarding the statistical analyses, presentation, and some other minor comments, all specified below.
My main concern with the MS is that it focuses so completely on showing that thalassochory happens in this fig species that it does not seem to even consider other possibilities for seed dispersal and gene flow among populations. Is it not possible that pollinators transport pollen between populations (line 451 specifically states so), and that this contributes to gene flow? Is it not possible that seeds are transported by birds or other animals, and that this founded populations and perhaps contributes to gene flow? Even if ocean currents are the main mode of seed dispersal, would not animals be involved in getting the figs to and from the water? How could you distinguish between the three hypotheses 1) thalassochory, 2) pollen dispersal, 3) animal seed dispersal? Could all three be happening, or could have happened in the past? How can you tell? Can you look into wind directions (for wasp dispersal)? Are animals/birds known to move between these sites? Are animals/birds known to eat these figs? I think the MS would be stronger if all these three possibilities were discussed, and brought up already in the introduction. The conclusion might be that you cannot for certain say that seeds were dispersed via ocean currents instead of by animals (birds included), but you can show that it would be possible, and that is still interesting and worthy of publication.
I think it would help the reader to present a bit more information about this fig species already in the introduction. Use an additional map to show current known distribution, indicate general wind directions and current direction already there. Give more details on the habitat – how close to the sea shore do they grow? Is suitable habitat available elsewhere in the area (but the Mearns fig not present), or is suitable habitat not present elsewhere because of human land use or other reasons? Also mention population sizes already in the introduction?
Figure 1 is very informative. How do you imagine the mature figs reach the sea water? How do you imagine the floating figs reaching land from the water? Do waves splash over the rocks in storms? Or do birds pick figs to eat and drop some in the water? Or perhaps pick up floating figs and eat them? Please give more information, or your best guess to what is happening. If the figs grow some way inland, seed dispersal by water could not be the only means, right?
Line 113 Mention here that you kept syconia in the water until all had sunk.
Line 120 How many seeds per fig were tested? Can you put this more detailed information in a supplement?
Line 121 What proportion of syconia had not developed fully? What proportion lacked seeds? What seemed to be the reason for lacking seeds?
Table 1
Make east Taiwan, south Taiwan and outlying islands in italics or such, and remove the summary N for those (60, 90, 60). I first thought those too were sampling locations.
Figure 2. It is clear here that variances are quite unequal among the groups, especially for MGT, so the data probably does not meet the assumptions of the ANOVA and Fisher LSD. Please provide more details on how you dealt with this (transformations?). It is likely that a different analysis than ANOVA would be more suitable for MGT and GRI. For analyzing proportions, typically a GLM with binomial errors is used. Explain in the methods what GRI is.
Table 3. Either explain MGT, GRI and FGP already in the legend, or use a superscript to refer to below the table.
Lines 226-232. Rephrase this because LSD is a posthoc to the ANOVA, so the results are intrinsically linked to each other. But likely you will end up using different tests anyway.
Figure 6. Very informative, great that you include the current vectors. Also please explain the current speed (in the text is fine). How far would a floating fig get in a day? In 10 days?
Discussion:
Point 4.1 See my earlier comments on how figs would get to and from the water. Is it not likely that animals/birds are involved here?
Line 348 Show the distribution on a map early on. Mention already in intro that you have sampled all known populations.
Lines 408-415. But are not those previously studied dioecious species growing in the forest understory, where there is little wind? That is a very different situation from your coastal figs. In a way, your coastal figs, even though small shrubs, are more like canopy species in that they are constantly exposed to wind and would allow easy wind dispersal for the wasps, no?
This thinking also applies to line 447.
Minor language points (in the interest of time I do not write out the specifics, but if you read these lines carefully I think you will understand what I mean).
Line 26 add of
Line 40 check grammar
Line 44 unclear
Line 46-47 unclear
Line 80 grammar
Line 116 unclear
Discussion – check language and grammar carefully throughout.
Lines 402-405. Please rephrase for clarity.
Comments on the Quality of English Language
See the review above
Author Response
Comments in the uploaded file.

Reviewer 2 Report
Comments and Suggestions for Authors
I like this manuscript. It presents evidence for the first time regarding the dispersion/colonization of the Ficus genus through oceanic. While the sampling could have been more thorough, I still believe the work is good and interesting. The results and discussion are solid and appropriately articulated. Consequently, I suggest its publication in this journal after minor revisions.
About the title: “First occurrence of thalassochory in the Ficus genus: Seed dispersal using the Kuroshio oceanic current” I do not think this is the first occurrence of thalassochory in the Ficus genus, probably it is the first EVIDENCE that this occurs. I would change the title to “First evidence of thalassochory in the Ficus genus: Seed dispersal using the Kuroshio oceanic current”
Abstract: Lines 18-19: I suggest deleting this sentence: “but living close to the sea can be deleterious in the current context of climate change.”
Line 44: “seeds” instead “seed”
Line 80: “hace colonized” instead of “have colonize”
Line 332: “much higher” instead of “very higher”
Line 335 “did not” instead of “didn’t”
Lines 402-404: Hard to follow. Please rewrite this sentences.
Figure 1: The sampling locations are barely visible in the map, I suggest increase the size” colour of the spots or add a new map.
469-471: I suggest deleting this sentence from the conclusions. It may fit better in the discussion.
I have reviewed the references and most seem fine, nevertheless there are some differences in format, I suggest authors review them carefully.
Author Response
Comments in the uploaded files
